# Behavior of Rotary Ultrasonic Machining of Ceramic Materials at a Wide Range of Cutting Speeds

**Marcel Kuruc ***[ID] **and Jozef Peterka**[ID]

Faculty of Materials Science and Technology, Slovak University of Technology in Bratislava, Vazovova 5, 812 43 Bratislava, Slovakia; jozef.peterka@stuba.sk
* Correspondence: marcel.kuruc@stuba.sk

**Abstract:** The paper deals with the behavior of the rotary ultrasonic machining process at different cutting speeds of ceramic materials. This process is relatively new; therefore, there are gaps in information about its behavior at near-critical parameters. We adjusted cutting speeds 10 times lower and 10 times higher than the recommended one. The observed parameters were machine load, tool wear, and surface roughness. Alumina and zirconia ceramics were used as materials. The results will help with the optimization of the cutting parameters of the rotary ultrasonic machining process.

**Keywords:** rotary ultrasonic machining; cutting speed; parameter behavior; surface roughness

## 1. Introduction

Ultrasonic machining is used for many machining technologies, such as turning, drilling, milling, and grinding [1–6]. In our paper, we focus on the combination of milling ultrasonic machining technology and grinding technology with diamond wheels. This hybrid technology is classified as rotary ultrasonic machining (RUM) [7,8].

Rotary ultrasonic machining (RUM) is classified as a hybrid technology that combines conventional ultrasonic machining and grinding with diamond wheels [7,8]. Very high cutting speeds are used for grinding, much higher than for RUM. However, there are no guidelines for cutting speeds when applying RUM. High-speed rotary ultrasonic machining is essentially the same as RUM and is similar to grinding with diamond wheels.

In cutting theory, vibration is one of the causes of roughness of the machined surface, and various methods have been devised to eliminate or at least reduce it to an acceptable level [9]. In contrast, RUM uses tool oscillation, which is controlled and, in this case, beneficial for the cutting process [10]. RUM can be used for machining various materials but is especially suited for hard and brittle materials [11–13]. In our paper, we machined alumina and zirconium ceramics with an emphasis on a high range of cutting speeds.

There have been several experiments performed where the influence of cutting parameters was observed. Pei et al. investigated cutting forces for RUM of brittle materials. They discovered the following tendencies of input variables for cutting forces: cutting forces decrease with increased cutting speed and increase with increased feed rate [14].

Material removal rate, cutting force, and edge-chipping were investigated by Pei et al. They studied the rotary ultrasonic machining of ceramic matrix composites by comparing RUM and diamond drilling. They learned that RUM provides a much lower cutting force than diamond drilling. A higher material removal rate is reached during RUM, as well. However, edge-chipping can occur at the entrance and exit of the holes during RUM. Fortunately, it can be reduced by suitable adjustment of machining parameters, such as maintaining a higher spindle speed and a lower feed rate [15].

Pei et al. investigated the RUM of potassium dihydrogen phosphate crystal, as well. According to their study, the surface roughness increases with increased cutting speed and feed rate. This behavior is different from the results reported for other materials,

where increasing the spindle speed might cause a vibration of the spindle and increase roughness [16].

Ya et al. analyzed the rotary ultrasonic machining mechanism. They considered the impact and grinding to be the main factors in the material removal rate for ultrasonic machining. They found three material removal mechanisms: impacting, abrasion, and ultrasonic cavitation. The biggest impacts on material removal rate were a static load, the grid of the abrasive, the concentration of the abrasive, the mechanical properties of the workpiece, tool material, cutting speed, and feed rate [17].

Wang et al. investigated the critical feed rate of effective RUM. They analyzed quartz glass and sapphire as workpiece materials. In their experiments, the cutting speed and feed rate were the variables. They discovered that increasing the cutting speed improves the critical feed rate and thereby elevates the effectiveness of the process [18].

Stoll and Neugebauer studied ultrasonic application in drilling. The cutting speed at the tool center is equal to zero during conventional drilling, and, therefore, the machining conditions worsen. However, when drilling is supported by ultrasound, the cutting speed at the tool center is different from zero, even if for a limited time, which improves the whole cutting process. Ultrasonic vibration affects the working angles of the tool by reducing friction on it and enhancing the chip formation. This increases the life of the tool [19].

A comprehensive review of rotary ultrasonic machining was made by Singh and Singhal. They analyzed 182 scientific articles and summarized the results. The behavior of adjusted parameters had a similar effect on observed parameters in different experiments. However, the adjusted values of the cutting parameters, as well as the machined material, often differ [20]. Another extensive review was provided by Singh, Ahuja, and Kapoor, where in addition to rotary ultrasonic machining they analyzed classic ultrasonic machining and chemical-assisted ultrasonic machining by studying 126 scientific articles [21]. Several research studies to determine cutting speed influence have been made, but none of them dealt with the importance of the cutting speed [22–33].

Our goal is to determine the behavior of selected ceramic materials when machined at very high and very low cutting speeds. This is the first article that utilizes such a high range of cutting speeds. The information we obtained will help to explain what happens when the upper or the lower limit of cutting speed is exceeded. This behavior will help in the optimization of cutting parameters in the RUM process.

## 2. Materials and Methods

### 2.1. Materials Used for the Experiments

We used two types of ceramic materials in the experiments—zirconia and alumina. Zirconia (zirconium dioxide, $ZrO_2$) is a white ceramic material with very low thermal conductivity. It is chemically unreactive and has superior thermal, mechanical, and electrical properties [34]. Unalloyed zirconia has a monoclinic crystal structure at room temperature; however, when it is stabilized by yttrium (Y) it has a metastable tetragonal structure [35]. The presence of the metastable phase enhances fracture toughness—when a higher amount of stress is present, the phase will change to the stable monoclinic phase, which absorbs part of the stress energy. This ability is called the transformation toughness [36]. Zirconium dioxide stabilized by yttrium is used especially in dental and medical applications [37]. Its properties are summarized in Table 1.

The properties of alumina are also summarized in the same table. Alumina (aluminum oxide (corundum), $Al_2O_3$) is considered the hardest oxide [38]. It has a hexagonal structure [39]. It is usually used as abrasive and is also found in cosmetics, electronics, chemistry, dentistry, etc. [40].

The workpieces were in block form with dimensions 45 mm × 45 mm × 20 mm for zirconia and 100 mm × 100 mm × 25 mm for alumina. According to Table 1, these oxidic ceramics have very different physical and mechanical properties. Therefore, different behaviors can be expected during their machining.

**Table 1.** Physical and mechanical properties of zirconia and alumina.

| Property | Unit | Zirconia | Alumina |
|---|---|---|---|
| Thermal expansion | $K^{-1}$ | $11 \times 10^{-6}$ | $8.4 \times 10^{-6}$ |
| Thermal conductivity | W/m.K | 2.5 | 30 |
| Melting point | °C | 2715 | 2072 |
| Density | $kg/m^{-3}$ | 5680 | 3980 |
| Fracture toughness | $MPa.m^{1/2}$ | 8 | 4.2 |
| Hardness HV | GPa | 12 | 22 |
| Young's modulus | GPa | 175 | 375 |

*2.2. Devices and Machines Used for the Experiments*

When high cutting speed is required, it can be achieved by high spindle speed, large cutting tool diameter, or their combination. In the experiments we used both high spindle speed (32,000 rpm) and the largest possible cutting tool diameter for the machine tool (Ø 30 mm). At such a high spindle speed, it was necessary to balance the cutting tool with the tool-holder. For this purpose, we used a Haimer Tool Dynamic 2009 balancing device. For balancing itself we used a pair of balancing rings made by Haimer labeled 79.530.32 Auswucht-Drehringe (Satz) für Schaft–Ø32. They were mounted on an Actor * HSK32 * D14h6 * High speed Max Rpm/40,000 ultrasonic tool-holder made by Sauer (DMG Mori, Stipshausen, Germany). On this tool-holder was also mounted a Schott 858009-3.25 6A9-Da30-3-8-14h6x8,4 MES3 D46H ultrasonic cutting tool made by Schott Diamantwerkzeuge (Stadtoldendorf, Germany).

The balanced tool set was mounted onto an Ultrasonic 20 linear machining center made by DMG Mori. This machine tool can continuously operate in five axes, and its spindle can rotate up to 40,000 rpm. Maximum spindle speed is usually limited to 10,000 rpm when ultrasound is active. However, due to the advanced tool-holder, it is possible to exceed this value. The machine tool is utilized for rotary ultrasonic machining of hard and brittle materials such as ceramics. During the machining process, it monitors the machine load of every axis, the load of the spindle, the torque of the cutting tool, and the performance of the ultrasonic generator. In our experiment these parameters were recorded continuously (according to the experimental set-up, there were 100 repetitions, therefore it was not necessary to record all observed parameters at once—only one parameter was observed at one repetition (to determine its behavior during the whole machining path), which made 12 possible repetitions for every observed parameter) but there were actually only six repetitions recorded for each of those parameters.

It was possible to measure the cutting tool dimensions (diameter, length) of this machine tool with the integrated BLUM Laser P87 tool probe. Differences in the length of the cutting tool were considered dimensional tool wear. The measurement was automatic and based on the interruption of the laser signal of the tool probe by the cutting tool during the tool´s rotation. When laser interruption was detected, this interruption was repeated three times. If the dispersion of measured values was too high, the measurement was automatically repeated or evaluated as unsuccessful. If the dispersion was low, the mean value of the tool length was recorded. The workspace with experimental set-up and balanced tool-holder with cutting tool are shown in Figure 1.

After the machining process, the surface roughness parameters were measured by the Mitutoyo SJ-210 roughness meter. The surface roughness was measured three times in the direction of the cutting tool movement (near the entrance, middle, and exit) and three times in the direction perpendicular to the cutting tool movement (near the entrance, middle, and exit). Values of roughness in both directions were very similar; therefore, the mean values of all six measurements for every cutting speed were recorded.

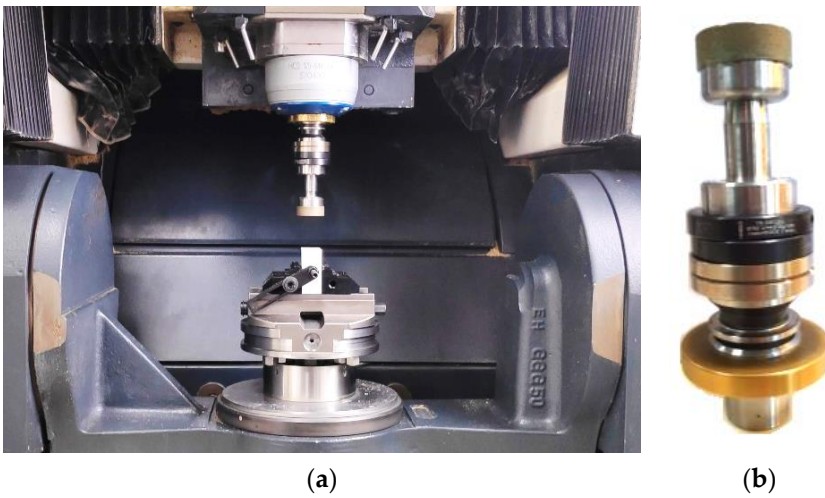

(**a**)                                                    (**b**)

**Figure 1.** Devices used: (**a**) Experimental set-up; (**b**) Balanced tool-holder.

*2.3. Cutting Conditions and Evaluated Parameters*

The cutting speed during the rotary ultrasonic machining process reached values of hundreds of meters per minute. Considering the tool size and maximum spindle speed, a reference cutting speed of 300 m/min was chosen. Two more speeds were used in the experiments. One of them was 10 times higher, the other was 10 times lower. This meant that the highest cutting speed was 100 times higher than the lowest one. If constant feed per revolution had been used, the cutting time for the lowest speed would be 100 times slower, which is impractical in practice. Moreover, because of the limited size of the workspace, the machine would not be able to achieve a feed rate 100 times higher than the lowest speed. Therefore, (due to practical and construction reasons), constant feed was not selected, but rather a feed rate of 1000 mm/min. The calculations of parameters are summarized in Table 2.

**Table 2.** Variable cutting parameters of the experiment.

| Parameter | Unit | Value | | |
|---|---|---|---|---|
| | | **Level 1** | **Level 2** | **Level 3** |
| Cutting speed | m/min | 30 | 300 | 3000 |
| Spindle speed | rpm | 320 | 3200 | 32,000 |
| Feed | mm | 3 | 0.3 | 0.03 |

We observed the cutting speed behavior with two different ceramic materials. Zirconia ($ZrO_2$) and alumina ($Al_2O_3$) were selected as workpieces. The axial depth of cut was constant at 0.01 mm. The radial step was 10 mm. These parameters were repeated 100 times (100 layers) and observed values were continuously recorded. The selection of constant machining parameters was based on the manufacturer's recommendation, literature review, and own experience [7,8]. We observed the machine loads across axes *X*, *Y*, *Z*, *A*, and *C* (%), the load of the spindle (%) (performance of the spindle was 4.5 kW), torque (Nm), and the performance of the ultrasonic generator (W) during the experiments; dimensional tool wear (tool length) before and after the process (mm); and finally surface roughness parameters (Ra, Rq, Rz) (μm).

*2.4. Achieved Values of Observed Parameters*

Monitoring of the observed parameters showed that despite the machining parameters used, the loads in axes *X*, *Y*, *A*, and *C* were unaffected. Only in the *Z* axis was the influence of cutting speed observed. Values of the relevant parameters for zirconia are recorded in Table 3, and those for alumina are recorded in Table 4.

**Table 3.** Recorded values of observed parameters during machining of zirconia.

| Cutting Speed [m/min] | Load in Z Axis [%] | Load of Spindle [%] | Torque [Nm] | Performance [W] |
|---|---|---|---|---|
| 30 | 17 | 45 | 3.0 | 13.5 |
| 300 | 5 | 11 | 0.9 | 11.8 |
| 3000 | 0 | 13 | 1.0 | 14.6 |

**Table 4.** Recorded values of observed parameters during machining of alumina.

| Cutting Speed [m/min] | Load in Z Axis [%] | Load of Spindle [%] | Torque [Nm] | Performance [W] |
|---|---|---|---|---|
| 30 | 12 | 11 | 0.94 | 13.2 |
| 300 | 0 | 8 | 0.57 | 12.0 |
| 3000 | 0 | 13 | 1.10 | 13.5 |

Parameters from Tables 3 and 4 were observed during the experiments. After the experiment, the dimensional tool wear was measured. The output measurement of one experiment was the input value for the next experiment. The measuring itself was provided by the NC program, where the dimensions were measured three times and the arithmetical value was recorded. The arithmetical lengths of the cutting tool after machining zirconia and alumina are recorded in Tables 5 and 6, respectively. The initial length was 112.730 mm. The removed volume of the machined material was constant for each workpiece material type: 450 mm$^3$ for zirconia and 1000 mm$^3$ for alumina. We also calculated the grinding ratio—the ratio between the material removed from the workpiece and the material removed (worn) from the cutting tool. The material removed from the workpiece was calculated according to workpiece length (45 mm for $ZrO_2$ and 100 mm for $Al_2O_3$), radial step (10 mm), and removed depth from the workpiece (0.01 mm $\times$ 100 layers = 1 mm). The material removed (worn) from the tool was calculated according to the cross-section of the tool (outer diameter 30 mm, inner diameter 24 mm) and the tool length decrement. A high grinding ratio value indicates that more material was removed from the workpiece, extending the tool life (bigger is better).

**Table 5.** Recorded values of tool length after machining of zirconia.

| Cutting Speed [m/min] | Cutting Tool Length [mm] | Length Difference [mm] | Grinding Ratio [-] |
|---|---|---|---|
| 30 | 112.707 | 0.023 | 76.9 |
| 300 | 112.703 | 0.002 | 408.4 |
| 3000 | 112.697 | 0.006 | 294.7 |

**Table 6.** Recorded values of tool length after machining of alumina.

| Cutting Speed [m/min] | Cutting Tool Length [mm] | Length Difference [mm] | Grinding Ratio [-] |
|---|---|---|---|
| 30 | 112.668 | 0.029 | 135.5 |
| 300 | 112.666 | 0.002 | 1980.8 |
| 3000 | 112.640 | 0.026 | 151.1 |

After machining the ceramic samples, the surface roughness parameters were measured. The measurements were repeated six times. Arithmetical surface roughness parameters Ra, Rq, and Rz after machining zirconia are recorded in Table 7, and the same arithmetical surface roughness parameters after machining alumina are recorded in Table 8.

**Table 7.** Recorded values of surface roughness after machining of zirconia.

| Cutting Speed [m/min] | Ra [μm] | Rq [μm] | Rz [μm] |
|---|---|---|---|
| 30 | 0.915 | 1.205 | 7.280 |
| 300 | 0.966 | 1.083 | 6.612 |
| 3000 | 0.321 | 0.466 | 3.570 |

**Table 8.** Recorded values of surface roughness after machining of alumina.

| Cutting Speed [m/min] | Ra [μm] | Rq [μm] | Rz [μm] |
|---|---|---|---|
| 30 | 1.100 | 1.534 | 11.271 |
| 300 | 1.397 | 2.312 | 18.738 |
| 3000 | 1.262 | 2.433 | 21.188 |

## 3. Results

### 3.1. Machine Loads Behavior

Graphs were created with the values of the observed parameters. Figure 2 shows the influence of cutting speed on load in the Z axis for both machined materials (zirconia and alumina). Figure 3 shows the influence of cutting speed on the load of the spindle for both machined materials. The influence of cutting speed on the torque of the cutting tool for both machined materials is shown in Figure 4, and the influence of cutting speed on the performance of the ultrasonic generator for both machined materials is shown in Figure 5.

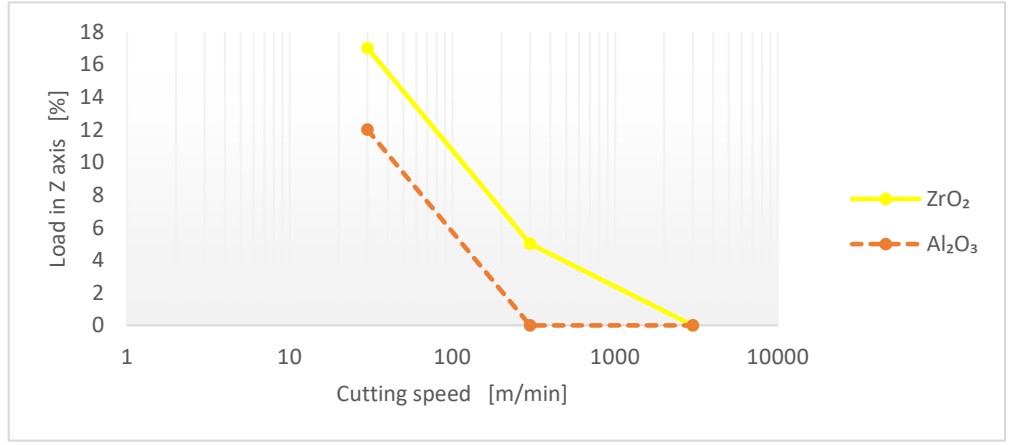

**Figure 2.** Influence of cutting speed on load in the Z axis.

According to Figure 2, Figure 3, Figure 4, Figure 5, the torque of the cutting tool showed nearly the same behavior as the spindle speed, which means there was a direct correlation. Most of the observed parameters decreased with increased cutting speed; only the performance of the ultrasonic generator was not affected by cutting speed or machined material. Increasing the cutting speed resulted in decreasing the machine load. This was caused by decreasing the feed per revolution—the cutting tool was cutting a lesser amount of material per revolution; therefore, it did not need a high cutting force. We can conclude that zirconia is more sensitive to low cutting speeds.

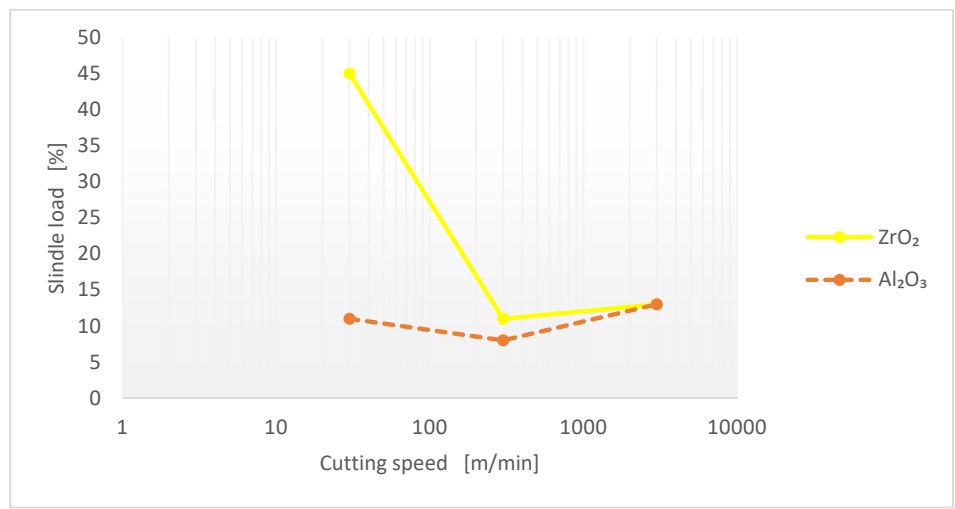

**Figure 3.** Influence of cutting speed on load of the spindle.

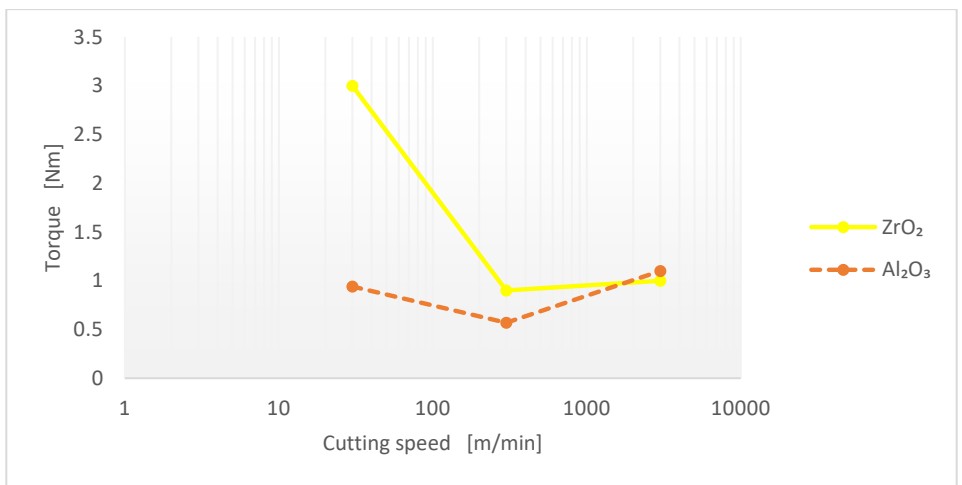

**Figure 4.** Influence of cutting speed on torque of the cutting tool.

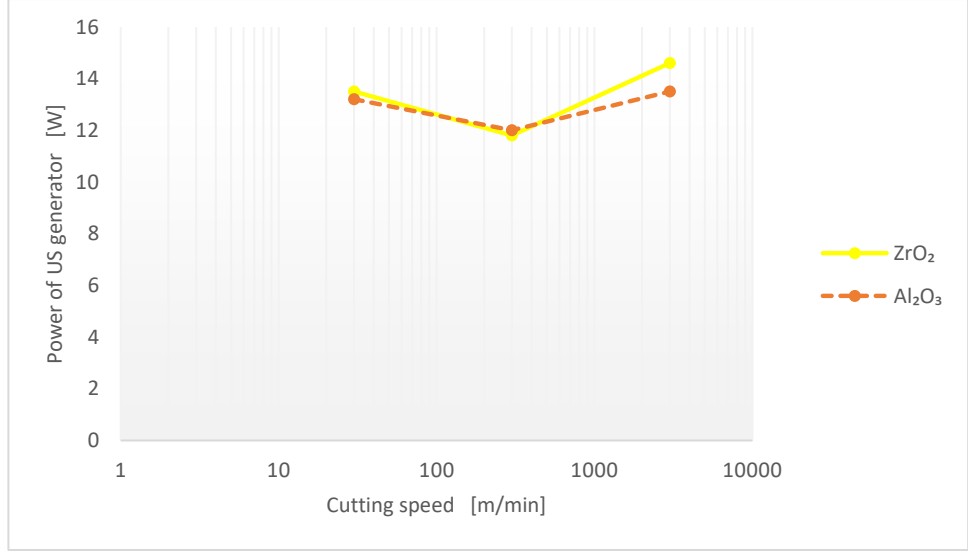

**Figure 5.** Influence of cutting speed on performance of the ultrasonic generator.

### 3.2. Tool Wear Behavior

The measurement of the dimensional tool showed the negative impact of low cutting speeds for both materials in comparison with their previous behavior. High cutting speed had a negative influence, but only for alumina, as shown in Figure 6. The very same cutting tool was used for both materials; therefore, the output length after the machining of zirconia was also the input length for the machining of alumina (having the same value).

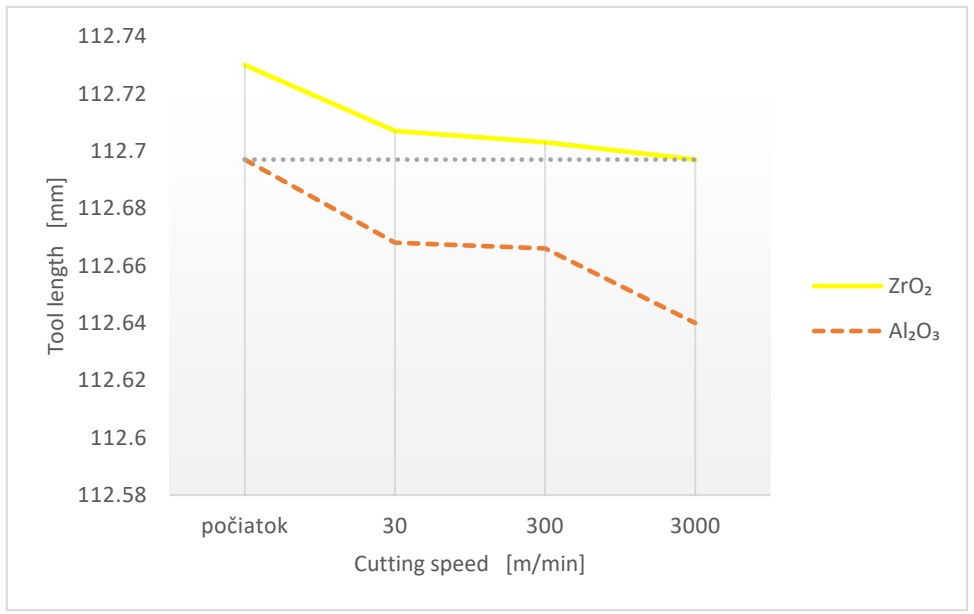

**Figure 6.** Influence of cutting speed on dimensional tool wear.

### 3.3. Surface Roughness Behavior

In terms of tool life, the standard cutting speed was an obvious choice for both materials. However, it was not so obvious in terms of surface roughness. As shown in Figure 7, at standard cutting speed we observed the worst impact on surface quality for both materials. Low surface roughness parameters can be reached by abrasive brushing; however, RUM produced components with complex shapes and at the same time reached high surface quality [41].

Other surface roughness parameters were similar in behavior to the Ra parameter. High cutting speed was beneficial in terms of the surface quality of zirconia. The reduction of machine loads is typical for this cutting speed as well.

In cutting theory, there is no theoretical dependence between the cutting speed and the roughness parameters of the machined surface. In contrast, we know from experimental observations that there is a range of cutting speeds where the increase in roughness is enormous, but with an increase in cutting speed beyond that range, up to the pace of high-speed machining, the surface roughness decreases in character. These conclusions have been observed for machining technologies with the so-called defined cutting-edge processes such as turning, drilling, milling [42–45]. In our research, we came to similar conclusions for RUM technology.

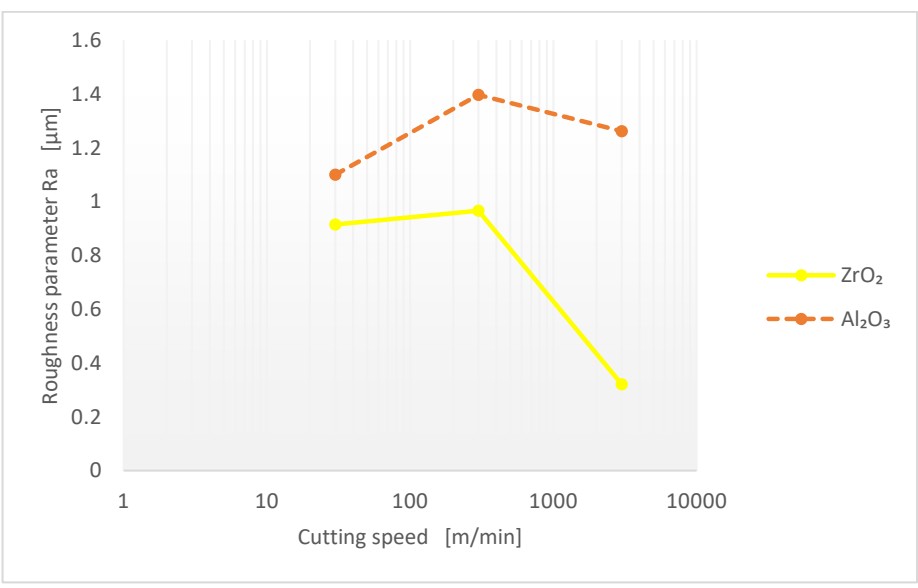

**Figure 7.** Influence of cutting speed on surface roughness parameter Ra.

## 4. Discussion

The results are summarized in Tables 9 and 10 for easier determination of cutting speed behavior. These tables are based on information from Tables 3 and 4, expanded by information from Table 5, Table 6, Table 7, Table 8. The performance of the ultrasonic generator was unaffected by cutting speed; therefore it is missing from these tables.

**Table 9.** Influence of cutting speed on significant observed parameters during machining of zirconia.

| Cutting Speed [m/min] | Load in Z Axis [%] | Load of Spindle [%] | Torque [Nm] | Ra [μm] | Tool Wear [μm] |
|---|---|---|---|---|---|
| 30 | 17 | 45 | 3.0 | 0.915 | 23 |
| 300 | 5 | 11 | 0.9 | 0.966 | 4 |
| 3000 | 0 | 13 | 1.0 | 0.321 | 6 |

**Table 10.** Influence of cutting speed on significant observed parameters during machining of alumina.

| Cutting Speed [m/min] | Load in Z Axis [%] | Load of Spindle [%] | Torque [Nm] | Ra [μm] | Tool Wear [μm] |
|---|---|---|---|---|---|
| 30 | 12 | 11 | 0.94 | 1.100 | 29 |
| 300 | 0 | 8 | 0.57 | 1.397 | 2 |
| 3000 | 0 | 13 | 1.10 | 1.262 | 26 |

Tables 9 and 10 summarize the results and present quantified values of the measured parameters. The surface roughness values were low—even the worst surface roughness would be acceptable in most applications. The load in the Z axis was often the most limiting parameter; however, its value was low in all cases. The torque of the tool was limiting, especially when cutting tools with low diameter were used. An ultrasonic milling cutter with a diameter of 30 mm was used in the experiments, which was robust enough. However, if a tool with a diameter of 4 mm had been used, it would be damaged at torque 3 Nm.

For easier evaluation of the behavior of the cutting speed and the subsequent decision on adjusting proper machining parameters, a linear transformation of regression function, interpolation, was applied to the values in Tables 9 and 10. The result of the linear transformation was the transformation of the values of one parameter into a range from −1 to +1, and all observed parameters are considered to be better with a lower value. Adding together all the values for one cutting speed indicates how suitable the parameter is (the

best is –5, the worst is +5). The results are recorded in Tables 11 and 12. In contrast with previous tables, the sum of the values after linear transformation was added in another column for easier evaluation of the cutting speed, considering all observed data.

**Table 11.** Linear transformation of cutting speed influence on observed parameters during machining zirconia.

| Cutting Speed [m/min] | Load in Z Axis [%] | Load of Spindle [%] | Torque [Nm] | Ra [μm] | Tool Wear [μm] | Sum [-] |
|---|---|---|---|---|---|---|
| 30 | 1 | 1 | 1 | 0.842 | 1 | 4.842 |
| 300 | −0.421 | −1 | −1 | 1 | −1 | −2.412 |
| 3000 | −1 | −0.882 | −0.905 | −1 | −0.789 | −4.576 |

**Table 12.** Linear transformation of cutting speed influence on observed parameters during machining alumina.

| Cutting Speed [m/min] | Load in Z Axis [%] | Load of Spindle [%] | Torque [Nm] | Ra [μm] | Tool Wear [μm] | Sum [-] |
|---|---|---|---|---|---|---|
| 30 | 1 | 0.200 | 0.396 | −1 | 1 | 1.596 |
| 300 | −1 | −1 | −1 | 1 | −1 | −3.000 |
| 3000 | −1 | 1 | 1 | 0.091 | 0.778 | 1.869 |

The lowest cutting speed provided the worst results for the machining of zirconia. The observed values were the worst in all cases except the surface roughness, where it achieved the second-to-worst value, similar to the worst value of the surface roughness for all cutting speeds for this workpiece material.

The results are based on the equal importance of all observed parameters. If one parameter were to be identified as more important than another (or vice versa), its value can be multiplied by a relevant factor. (For example, if surface roughness is not considered important, it can be multiplied by 0, and if dimensional tool wear is considered twice as important as other parameters, it can be multiplied by 2.)

It is necessary to mention that it was not only the cutting speed that affected the results. When high-speed cutting occurred at 3000 m/min, the high feed cutting occurred at 20 m/min, which reached a feed of 3 mm/rev. From the literature review, we can see that the feed has an opposite effect to cutting speed, meaning that the effect of cutting speed is amplified at a constant feed rate, which were the conditions of performed experiments. However, from the cutting theory viewpoint, feed per tooth should have a great impact on the surface roughness parameters of the machined surface; therefore, we would expect the lowest roughness at the lowest feed (i.e., the highest cutting speed), which was confirmed only for zirconia ceramic at higher speeds. We did not observe this behavior for zirconia at low cutting speeds or for alumina ceramic at the entire range of cutting speeds.

The different results for zirconia and alumina could be caused by their different physical and mechanical properties. Zirconia has very low thermal conductivity, while alumina has high (for a ceramic material) thermal conductivity. This property affects the composition of temperatures in the cutting zone—at high thermal conductivity, generated heat is drained by the workpiece as well, which results in a lower temperature of the cutting tool (this is especially important at higher cutting speeds). Alumina is much harder and more brittle than zirconia. It affects the machining process itself. The main removal mechanism in RUM is a brittle fracture, which means brittle materials are easier to machine. Zirconia is considered to be one of the toughest ceramic materials.

## 5. Conclusions

According to our results, we can conclude the following behavior with zirconia ceramic:

- High cutting speeds are suitable for rotary ultrasonic machining (RUM) of zirconia. The lowest machine loads and surface roughness, and very low spindle load and torque and tool wear were achieved.

- In contrast, low cutting speed (high feed per revolution) is not suitable for this workpiece material. The worst results of all considered parameters were observed.
- Medium (standard) cutting speeds achieve very good results as well—the lowest spindle load and torque and tool wear, low machine loads, and the worst surface roughness (similar to the one achieved at the lowest cutting speed) were reached; however, the value for parameter Ra was under 1 μm, which is still considered a smooth surface.

According to the results, we can conclude the following behavior with alumina ceramic:

- Considered as workpiece material, at medium cutting speeds alumina responded almost the same as zirconia—the lowest values of machine loads, spindle load, torque and tool wear, and the highest surface roughness, at a value of 1.4 μm for the Ra parameter (which can still be considered a smooth surface), were reached.
- On the other hand, very high or low cutting speeds do not suit this material. At high cutting speeds, high spindle load and torque occur. High machine loads and tool wear occur at low cutting speeds.

One potential risk exists at high cutting speeds. At higher cutting speeds, more heat is generated, which manifests as increased cutting temperature. The active part of ultrasonic tools consists of diamond grains, and diamond decays into graphite at elevated temperatures. This phenomenon was not observed in the performed experiments, but we should not underestimate the possibility in terms of preventing future failure during long-term usage.

We recommend the following machining parameters according to the abovementioned data:

- Medium to high cutting speeds are proper for machining of zirconia ceramics (e.g., 500 to 2000 m/min);
- Medium cutting speeds are proper for machining of alumina ceramics (e.g., 300 to 600 m/min);
- High feed is not suitable for zirconia or alumina.

The results can be applied to the optimization of cutting parameters for these materials. Further experiments should be focused on confirmation of the obtained behavior by using different cutting speeds. At present, it is also possible to use the methods of mathematical modeling and simulation methods for research into RUM processes [46,47]. Moreover, the effect of cutting speed on other (ceramic) materials should be examined as well.

**Author Contributions:** Conceptualization, M.K. and J.P.; methodology, M.K.; software, M.K.; validation, M.K.; formal analysis, M.K. and J.P.; investigation, M.K.; resources, J.P.; data curation, M.K.; writing—original draft preparation, M.K.; writing—review and editing, M.K. and J.P.; visualization, M.K.; supervision, J.P.; project administration, J.P.; funding acquisition, J.P. All authors have read and agreed to the published version of the manuscript.

**Funding:** This research was funded by a project of the VEGA grant agency of the Ministry of Education, Science, Research, and Sport of the Slovak Republic and Slovak Academy of Sciences, no. 1/0019/2020: "Accurate calculations, modeling, and simulation of new surfaces based on physical causes of machined surfaces and additive technology surfaces in machinery and robotic machining conditions".

**Acknowledgments:** The article was written with the support of a project of VEGA grant agency of the Ministry of Education, Science, Research and Sport of the Slovak Republic and Slovak Academy of Sciences, no. 1/0019/2020: "Accurate calculations, modeling, and simulation of new surfaces based on physical causes of machined surfaces and additive technology surfaces in machinery and robotic machining conditions", and the APVV Project of Slovak Research and Development Agency of the Ministry of Education, Science, Research, and Sport of the Slovak Republic, no. APVV-16-0057: "Research into the Unique Method for Treatment of Cutting Edge Microgeometry by Plasma Discharges in Electrolyte to Increase the Tool Life of Cutting Tools in Machining of Difficult-to-Machine Materials."

**Conflicts of Interest:** The authors declare no conflict of interest. The funders had no role in the design of the study; in the collection, analyses, or interpretation of data; in the writing of the manuscript, or in the decision to publish the results.

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
