# Peer review of "Behavior of Rotary Ultrasonic Machining of Ceramic Materials at a Wide Range of Cutting Speeds"

_machines, doi:10.3390/machines9080164_

Round 1

Reviewer 1 Report

Recommendation to authors:

The article need to be completed with following suggestion to bring higher quality of this paper

  1. In introduction - Goals of article are missing – what is new and what is different comparing with other articles
  2. Give some necessary material properties of used ceramics – mainly hardness, Material properties also influence the surface roughness creation  in RUM process
  3. Explain and describe how was Grinding ratio calculated in Table 4
  4. Explain what caused the fact that the cutting tool length after process was higher than the initial cutting tool length, Fig 7, when zirconia ceramic material was machined
  5. Explain and describe how was measured the tool wear
  6. Explain the surface roughness measurement direction
  7. Which type of linear transformation was used?

Author Response

Thank you for your comments and recommendations.

  1. We added the goals of the article in the introduction.

Lines 79 – 83:

Our goal is to determine the behavior of selected ceramic materials when ma-chined at very high and very low cutting speeds. This is the first article that utilizes such a high range of cutting speeds. The information we obtained will help to explain what happens when the upper or the lower limit of cutting speed is exceeded. This behavior will help in the optimization of cutting parameters in the RUM process.

  1. We added a subchapter about used materials.

Lines 85 – 107:

2.1. Materials used for the experiments

  1. We explained and described the grinding ratio calculation.

Lines 200 – 208:

We also calculated the grinding ratio—the ratio between the material removed from the workpiece and the material removed (worn) from the cutting tool. The material removed from the workpiece was calculated according to workpiece length (45 mm for ZrO2 and 100 mm for Al2O3), radial step (10 mm), and removed depth from the workpiece (0.01 mm x 100 layers = 1 mm). The material removed (worn) from the tool was calculated according to the cross-section of the tool (outer diameter 30 mm, inner diameter 24 mm) and the tool length decrement. A high grinding ratio value indicates that more material was removed from the workpiece, extending the tool life (bigger is better).

  1. Cutting tool length was after the process always lower. Output length after zirconia machining is input length for alumina machining.

     Lines 256 – 258:

     The very same cutting tool was used for both materials; therefore, the output length after the machining of zirconia was also the input length for the machining of alumina (having the same value).

  1. We explained and described tool wear measurements.

Lines 133 – 140:

It was possible to measure the cutting tool dimensions (diameter, length) of this machine tool with the integrated BLUM Laser P87 tool probe. Differences in the length of the cutting tool were considered dimensional tool wear. The measurement was automatic and based on the interruption of the laser signal of the tool probe by the cutting tool during the tool´s rotation. When laser interruption was detected, this interruption was repeated three times. If the dispersion of measured values was too high, the measurement was automatically repeated or evaluated as unsuccessful. If the dispersion was low, the mean value of the tool length was recorded.

  1. We explained the surface roughness measurement direction.

Lines 148 – 153:

After the machining process, the surface roughness parameters were measured by the Mitutoyo SJ-210 roughness meter. The surface roughness was measured three times in the direction of the cutting tool movement (near the entrance, middle, and exit) and three times in the direction perpendicular to the cutting tool movement (near the entrance, middle, and exit). Values of roughness in both directions were very similar; therefore, the mean values of all six measurements for every cutting speed were recorded.

  1. We determined the type of used linear transformation.

Lines 305 – 307:

For easier evaluation of the behavior of the cutting speed and the subsequent decision on adjusting proper machining parameters, a linear transformation of regression function – interpolation. 

Reviewer 2 Report

A very good paper devoted to the new progressive and prespective technology, for many applications even the only only suitable.

Extensive experimental work, broad intervals of experiamntal conditions, a  good navigation of readers understand the scientific background and also to employ the results and conclusions.

A unique combination of fundamental research and practical use in wide industrial practice. I fully recommend to publish it as it is.

1.The paper deals mainly with the rotary ultrasonic machining process of ceramic materials. The workpiece was used alumina and zirconia ceramics, possibly the most interesting ceramics at all. 

2. This process is quite new and therefore there lack of information about the technology today. 

3. The complexity of the paper: observed parameters are machine load, tool wear and surface roughness. 

4. To quantify the loading of the spindle in watts regarding the methodology

5.The conclusions consistent with the evidence and arguments presented and they address the main question posed

6. the references are appropriate

7. I miss statistics: means, dispersions, the limits of confidence intervals, limits a statistical behaviour of the variables. Also an analysis of waviness of the surfaces might be interesting.

Author Response

Thank you for your comments and recommendations.

  1. You are right, thank you.
  2. You are right.
  3. Thank you.

  1. We added spindle performance in watts.

Lines 176 – 180:

We observed the machine loads in axes X, Y, Z, A, C (%), the load of the spindle (%) (performance of the spindle was 4.5 kW), torque (Nm), and the performance of the ultrasonic generator (W) during the experiments, and dimensional tool wear (tool length) before and after the process (mm), and finally surface roughness parameters (Ra, Rq, Rz) (µm).

  1. Thank you.
  2. Thank you.

  1. Extended statistics should undoubtedly be interesting, but we did not have enough time for that or. we solved the pilot research, where we focused on a wide range of cutting speeds that would set the direction of further research for us (as well as other researchers). We will think about it in further experiments.

Reviewer 3 Report

In my opinion authors should correct/clarify the following issues:

#1) In the title authors state that the paper deals with the behavior “at the high range of cutting speeds”. However, only three cases are analysed in the paper. In addition, in my opinion author should include in the title that the study is applied to ceramic materials.

#2) According to data included in the paper, machines can rotate up to 40000 rpm and maximum spindle speed is limited to 10000 rpm. Why authors have not considered a higher number of cases of “high speed” as they state in the paper. In addition, in my opinion authors should include in tables and figures results for more cutting speeds since three cases (30, 300 and 3000 m/min)  seems to be a few in order to reveal the evolution of analyzed variables.

#3) Line 97. For the sake of clarity, authors should include figures and a deep explanation of test set up instead of commercial figures of the test machines. In addition, they should describe deeply how all the test parameters analyzed in the paper were obtained.

#4) Line 186. All the figures show in the x-axis seems to be in log scale but it is not shown in the plots. Please show all the figures 3-8 properly and in addition label the x-axis in log scale properly identifying 1, 10, 100, 1000 and so on instead of 30, 300 and 3000.

#5) Line 197.200. Please provide a properly discussion for Figs 3 to 6 since the discussion included in the paper is so general and short.

#6) Please remove the references from [29] to [35] since there are just links to catalogs.

#7) The reference [31] is missed in the manuscript, it is not commented.

Minor changes:

Please use subindex for variables such as R1, Rq, Rz and for Al2O3 and ZrO2 in plots.

Author Response

Thank you for your comments and recommendations.

  1. We modified the title.

Lines 2 – 3:

Behavior of rotary ultrasonic machining of ceramic materials at the high range of cutting speeds.

  1. You are right. But we solved the pilot research, where we focused on a wide range of cutting speeds that would set the direction of further research for us (as well as other researchers).

  1. We replaced commercial figures with real pictures of the experimental setup. We described how analyzed parameters were obtained.

Line 143.

Lines 125 – 140, 148 – 153:

During the machining process, it monitors the machine load of every axis, the load of the spindle, the torque of the cutting tool, the performance of the ultrasonic generator. In our experiment these parameters were recorded continuously (according to the experimental set-up, there were 100 repetitions, therefore it was not necessary to record all observed parameters at once—only one parameter was observed at one repetition (to determine its behavior during the whole machining path), which made 12 possible repetitions for every observed parameter) but there were actually only 6 repetitions recorded for each of those parameters.

It was possible to measure the cutting tool dimensions (diameter, length) of this machine tool with the integrated BLUM Laser P87 tool probe. Differences in the length of the cutting tool were considered dimensional tool wear. The measurement was automatic and based on the interruption of the laser signal of the tool probe by the cut-ting tool during the tool´s rotation. When laser interruption was detected, this interruption was repeated three times. If the dispersion of measured values was too high, the measurement was automatically repeated or evaluated as unsuccessful. If the dispersion was low, the mean value of the tool length was recorded.

After the machining process, the surface roughness parameters were measured by the Mitutoyo SJ-210 roughness meter. The surface roughness was measured three times in the direction of the cutting tool movement (near the entrance, middle, and exit) and three times in the direction perpendicular to the cutting tool movement (near the entrance, middle, and exit). Values of roughness in both directions were very similar; therefore, the mean values of all six measurements for every cutting speed were recorded.

  1. We corrected the labeling of the log scale of the plots.

Lines 232, 235, 238, 241, 270.

  1. We extended the discussion for Figs 3 to 6.

Lines 244 – 251:

According to Figures 2 to 5, the torque of the cutting tool showed nearly the same behavior as the spindle speed, which means there was a direct correlation. Most of the observed parameters decreased with increased cutting speed; only the performance of the ultrasonic generator was not affected by cutting speed or machined material. Increasing the cutting speed resulted in decreasing the machine load. This was caused by decreasing the feed per revolution—the cutting tool was cutting a lesser amount of material per revolution; therefore, it did not need a high cutting force. We can conclude that zirconia is more sensitive to low cutting speeds.

  1. We removed the references from [29] to [35].

  1. The reference [31] was placed in the sentence about tool-holder (previously Line 87).

Lines 116 – 117:

They were mounted on an Actor * HSK32 * D14h6 * High speed Max Rpm / 40,000 ultrasonic tool-holder made by Sauer (DMG Mori).

  1. We added subindexes for materials in plots.

Lines 232, 235, 238, 241, 259, 270.

Reviewer 4 Report

Rotary ultrasonic machining (RUM) was invented by DMG, however you ought to aim at real common knowledge, for instance follow the points below:

  • Half of your device pictures are to be eliminated, readers do not care about the toolholder balance bench…etc. The only left must be the DMG machine and toolholder (in one figure). Usually young researcher include all systems, and several are commercial ones without a direct influence on the results. Did you have different results with other balance machine…my guess is no
  • Results are interesting because ceramics are usually processed with RUM, in all exhibition fairs the DMG machines are showing those examples, that makes very important to include a real literatures review. I do not see works by Celaya, Pujana, Polvorosa, or Journal of Materials Engineering and Performance 25 (11), 5076-5086…really there were people working with machining and US together. Ultrasonically-assisted machining (UAM), which uses a specially designed piezoelectric transducer working in tandem with a traditional turning, drilling or milling machine. Was proposed by Babitsky of Loughborough
  • Conclusions, I should like to read them as points, one per each idea and claim.
  • You showed variation between zirconia and alumina, however the discussion about why they are so different is weak, use some ideas from ceramic works, give some view of structures, grains, etc…the difference must be explain.
  • How many repetition did you check at each experiments?
  • Eliminate 29-35 web pages references. This is very ugly, a paper is not a Msc with the usual students things.
  • Cutting Edge Preparation: Ok this is interesting.
  • You did not check The International Journal of Advanced Manufacturing Technology  or International Journal of Mechatronics and Manufacturing Systems

You see above that with current version decision must be Major, or even reject. However the application is really interesting and I encourage you to improve it.

Author Response

Thank you for your comments and recommendations.

  1. We eliminated several pictures of devices.

  1. We added recommended references.

Lines 19 – 22, 418 – 432:

Ultrasonic machining is used for many machining technologies, such as turning, drilling, milling, and grinding [1,2,3,4,5,6]. In our paper, we focus on the combination of milling ultrasonic machining technology and grinding technology with diamond wheels. This hybrid technology is classified as rotary ultrasonic machining (RUM) [7,8].

  1. Celaya, A.; Campa, F.J.; de Lacalle, LNL.; Marina, D. The Effects of Ultrasonic Vibration Parameters on Machining Performance in Turning of Mild Steels. Int. Conf. on Advances in Materials and Processing Technologies, PTS ONE AND TWO. AIP Conference Proceedings 1315, 1139-1144, 2010.
  2. Celaya, A.; Pujana, J.; Lopez De Lacalle, L.N.; Rivero, A.; Campa, F.J.]. Improvement of turning and drilling by ultrasonic assistance. DAAAM Int. Scien. Book 2008, pp. 205-2018.
  3. Celaya, A.; de Lacalle, LNL.; Campa, F.J.; Lamikiz, A.; Ultrasonic Assisted Turning of mild steels. Int. J. of Materials and Product Technology. 2010, vol 37, Iss 1-2, pp 60-70, DOI 10.1504/IJMPT.2010.029459.
  4. Maurotto, A.; Roy, A.; Babitsky, V.I.; Silberschmidt, V.V.; Analysis of machinability of Ti- and Ni-based alloys. Advanced Materials and Structures IV. Book Series: Solid State Phenomena. Vol. 188 pp. 330-338, 2012, DOI: 10.4028/www.scientific.net/SSP.188.330.
  5. Li, X.; Meadows, A.; Babitsky, V.; Parkin, R.; Experimental analysis on autoresonant control of ultrasonically assisted drilling. Mechatronics. Vol. 29 p.57-66, 2015, DOI: 10.1016/j.mechatronics.2015.05.006.
  6. Suárez, A.; Veiga, F.; López de Lacalle, L.N.; Roberto Polvorosa, R.; Lutze, S.; Wretland, A. Effects of ultrasonics-assisted face milling on surface integrity and fatigue life of Ni-Alloy 718. J. of Mat. Engineering and Performance. Vol. 25 Iss. 11 pp.5076-5086.

  1. We added points in the conclusion.

Lines 350 – 385:

According to our results, we can conclude the following behavior with zirconia ceramic:

  • High cutting speeds are suitable for rotary ultrasonic machining (RUM) of zirconia. The lowest machine loads and surface roughness, and very low spindle load, torque and tool wear were achieved.
  • In contrast, low cutting speed (high feed per revolution) is not suitable for this workpiece material. The worst results of all considered parameters were observed.
  • Medium (standard) cutting speeds achieve very good results as well—the lowest spindle load, torque and tool wear, low machine loads, and the worst surface roughness (similar to the one achieved at the lowest cutting speed) were reached; however, the value for parameter Ra was under 1 µm, which can still be considered a smooth surface.

According to the results, we can conclude the following behavior with alumina ceramic:

  • Considered as workpiece material, at medium cutting speeds alumina responded almost the same as zirconia—the lowest values of machine loads, spindle load, torque and tool wear, and the highest surface roughness, at a value of 1.4 µm for the Ra parameter (which can still be considered a smooth surface), were reached.
  • On the other hand, very high or low cutting speeds do not suit this material. At high cutting speeds, high spindle load and torque occur. High machine loads and tool wear occur at low cutting speeds.

One potential risk exists at high cutting speeds. At higher cutting speeds more heat is generated, which manifests as increased cutting temperature. The active part of ultrasonic tools consists of diamond grains, and diamond decays into graphite at elevated temperatures. This phenomenon was not observed in the performed experiments, but we should not underestimate the possibility in terms of preventing future failure during long-term usage.

We recommend the following machining parameters according to the abovementioned data:

  • Medium to high cutting speeds are proper for machining of zirconia ceramics (e.g., 500 to 2000 m/min);
  • Medium cutting speeds are proper for machining of alumina ceramics (e.g., 300 to 600 m/min);
  • High feed is not suitable for zirconia or alumina.

  1. We explained the difference of variation between zirconia and alumina.

Lines 340 – 348:

The different results for zirconia and alumina could be caused by their different physical and mechanical properties. Zirconia has very low thermal conductivity, while alumina has high (for a ceramic material) thermal conductivity. This property affects the composition of temperatures in the cutting zone—at high thermal conductivity, generated heat is drained by the workpiece as well, which results in a lower temperature of the cutting tool (this is important especially at higher cutting speeds). Alumina is much harder and more brittle than zirconia. It affects the machining process itself. The main removal mechanism in RUM is a brittle fracture, which means brittle materials are easier to machine. Zirconia is considered to be one of the toughest ceramic materials.

  1. We quantified the number of repetitions.

Lines 125 – 140, 148 – 153:

During the machining process, it monitors the machine load of every axis, the load of the spindle, the torque of the cutting tool, the performance of the ultrasonic generator. In our experiment these parameters were recorded continuously (according to the experimental set-up, there were 100 repetitions, therefore it was not necessary to record all observed parameters at once—only one parameter was observed at one repetition (to determine its behavior during the whole machining path), which made 12 possible repetitions for every observed parameter) but there were actually only 6 repetitions recorded for each of those parameters.

It was possible to measure the cutting tool dimensions (diameter, length) of this machine tool with the integrated BLUM Laser P87 tool probe. Differences in the length of the cutting tool were considered dimensional tool wear. The measurement was automatic and based on the interruption of the laser signal of the tool probe by the cutting tool during the tool´s rotation. When laser interruption was detected, this interruption was repeated three times. If the dispersion of measured values was too high, the measurement was automatically repeated or evaluated as unsuccessful. If the dispersion was low, the mean value of the tool length was recorded.

After the machining process, the surface roughness parameters were measured by the Mitutoyo SJ-210 roughness meter. The surface roughness was measured three times in the direction of the cutting tool movement (near the entrance, middle, and exit) and three times in the direction perpendicular to the cutting tool movement (near the entrance, middle, and exit). Values of roughness in both directions were very similar; therefore, the mean values of all six measurements for every cutting speed were recorded.

  1. We eliminated 29-35 web pages references.

  1. Thank you.

  1. We checked recommended and relevant International Journals.

We have reviewed the recommended and relevant international journals. We found interesting and current references in the IJAMT magazine and added them to the list of literature [47, 48].

Round 2

Reviewer 1 Report

There are no comments to authorsconcerning the content of article. Recommendation:  English needs professional  proofreading.

Author Response

Thank you very much for your review. The professional proofreading of the English has been already made by MDPI itself.

Author Response

Thank you very much for your review.

Reviewer 4 Report

Good paper

Author Response

Thank you very much for your review.